# Information sharing practices during the COVID-19 pandemic: A case study about face masks

**Hannah Baker**[1]*, **Shauna Concannon**[1], **Emily So**[2]

**1** Centre for Research in the Arts, Social Sciences and Humanities (CRASSH), University of Cambridge, Cambridge, United Kingdom, **2** Department of Architecture, University of Cambridge, Cambridge, United Kingdom

* heb51@cam.ac.uk

## Abstract

This article contributes an empirical analysis of information sharing practices on Twitter relating to the use of face masks in the context of COVID-19. Behavioural changes, such as the use of face masks, are often influenced by people's knowledge and perceptions, which in turn can be affected by the information available to them. Face masks were not recommended for use by the UK public at the beginning of the COVID-19 pandemic. Due to developments in scientific understanding, the guidance changed and by the end of 2020 they were mandatory on public transport and in shops. This research examines tweets in this longitudinal context and, therefore, provides novel insights into the dynamics of crisis communication in an ongoing crisis event with emerging scientific evidence. Specifically, analysis of the content of tweets, external resources most frequently shared, and users sharing information are considered. The conclusions contribute to developing understanding of the digital information ecology and provide practical insights for crisis communicators. Firstly, the analysis shows changes in the frequency of tweets about the topic correspond with key guidance and policy changes. These are, therefore, points in time official channels of information need to utilise the public's information seeking and sharing practices. Secondly, due to changes in face mask guidance and policy, the current literature on digital information ecology is insufficient for capturing the dynamic nature of a long-term ongoing crisis event. Challenges can arise due to the prolonged circulation of out-of-date information, i.e. not strategic misinformation, nor "mis"-information at all, which can have serious ramifications for crisis communication practitioners. Thirdly, the role of traditional media and other journalism/broadcasting platforms in shaping conversations is evident, as is the potential for scientific organisations' and individual people's Twitter user accounts. This plurality of contributors needs to be acknowledged and understood to inform crisis communication strategies.

**Data Availability Statement:** All relevant data are within the paper and its Supporting Information files. Please note, full Tweets have not been provided due to Twitter copyright permissions. However, we have provided Tweet IDs in the

supporting information and the External URLs analysed.

**Funding:** Dr Hannah Baker's and Dr Emily So's research project is called 'Expertise Under Pressure', and Dr Shauna Concannon's is entitled 'Giving Voice to Digital Democracies'. Both projects are part of the Centre for Research in the Arts, Social Sciences and Humanities (CRASSH) at the University of Cambridge, and funded by the Humanities & Social Change International Foundation (https://hscif.org/cambridge/). The funders had no role in study design, data collection and analysis, decision to publish, or preparation of the manuscript.

**Competing interests:** The authors have declared that no competing interests exist.

## Introduction

On 31 December 2019, an unknown cause of pneumonia was reported to the World Health Organisation (WHO) county office in China [1]. Subsequently the outbreak of this new disease, now known as COVID-19, led to the WHO declaring it a global pandemic on the 11 March 2020 [2]. Mitigation measures to reduce the spread were implemented throughout the world, with '*hands, face, space*' being one of the slogans introduced in September 2020 by the UK Government to promote the non-pharmaceutical mitigation measures; meaning: wash your hands, wear a face mask (also referred to as face covering) and keep your space from people outside your own household [3]. For the ease of reading, the term 'face masks' is used throughout this paper to refer to both face masks and face coverings for public use e.g., a non-medical mask such as cloth rather than a respirator.

The effectiveness of mitigation measures, such as face mask use, is reliant on strong public support and the masses complying [4–6]. People's level of knowledge, attitudes and risk perceptions can determine their readiness to accept these types of behavioural changes [4, 7]. As individual perceptions of threats are often influenced by the information received [8], research has demonstrated that clear, consistent and effective public messaging is likely to increase the public's adherence to guidelines [9, 10]. This study focuses on one of the recommended mitigation strategies: face masks. Using a Twitter dataset (study period: 22 January—1 August 2020), the dissemination of information about face masks is analysed, with the aim of developing an understanding on the digital information ecology and how these information sharing practices could have affected the public's understanding of the scientific evidence and expertise related to their use. These insights can be used to guide future public information campaigns [11]. This aim is underpinned by the following research questions (RQ):

**RQ1:** What type of information about face masks was deemed as newsworthy and shareable?

**RQ2:** When did users choose to share information about face masks during the COVID-19 pandemic?

**RQ3:** What were the main external resources of information about face masks?

**RQ4:** Which user's tweets were most widely circulated?

The aim of the first question is to set the scene and identify what topics were discussed and whether scientific evidence and expertise formed part of this. The other questions explore the complexity of information circulation practices. As we are interested in the circulation of scientific evidence, we also investigate if these information sharing behaviours differ for scientific content.

We begin by outlining the changes in the guidance and use of face masks in the UK and its devolved nations. The academic foundation for this work is then discussed and we set the theoretical context by drawing upon work and theories associated with the digital information ecology and crisis communication. The methodology section provides detail on the Twitter datasets we have used for the analysis, including how the tweets were filtered and then analysed. Our findings are presented under the headings: topic modelling and URL subject content; frequency of tweets and circulation of scientific resources; popular domains; and user profiles with the most interaction. These findings are then discussed under headings corresponding with the research questions. Our conclusions and suggested further work are presented at the end of the paper.

Analysing tweets in the longitudinal context of an ongoing crisis such as COVID-19 provides novel insight and understanding into the dynamics and influences of crisis communication [8]. The research, therefore, contributes to developing understanding of the digital

information ecology, in particular the circulation of scientific information and expertise in an ongoing crisis context which is subject to uncertainty and emerging scientific evidence. This includes insight into when information is shared, including concerns about the recirculation of out-of-date articles, and the plurality of information resources and communicators online in information circulation, including the dominant role of traditional media but also some scientific organisations and individuals.

## Changes in the guidance and use of face masks

Hand washing and physical distancing were encouraged since the introduction of the UK's COVID-19 restrictions in March 2020, however, the use of face masks was not mandatory, or even recommended by the Government and its devolved nations until a later date. The first devolved nation to recommend use on public transport was Scotland on 28 April 2020, whilst the first nation to mandate use on public transport was England from 15 June 2020. These requirements were followed by additional restrictions including the use of masks in shops [12]. As shown in Table 1, the changes in guidelines and mandating masks varied between the devolved nations during 2020.

The purpose of using face masks for the public is to reduce source virus transmission, particularly from asymptomatic or pre-symptomatic individuals when physical distancing is not possible or predictable [9, 25]. The Scientific Advisory Group for Emergencies (SAGE) and its specialist sub-committees provide advice for the UK Government [26]. Near the start of the then epidemic (4 February 2020, before COVID-19 was classified as a global pandemic by the WHO), SAGE was advised by the New and Emerging Respiratory Virus Threats Advisory Group (NERVTAG) that there was '*limited to no evidence of the benefits of the general public wearing facemasks as a preventative measure*' and face masks were only advised for health and social care workers visiting people who may be infectious, whilst surgical face masks were recommended for symptomatic individuals [27]. As more studies were conducted on COVID-19, the scientific evidence developed [10]. Minutes from a meeting of SAGE on 21 April 2020 (released 29 May 2020), states the evidence of masks for source control and for protecting the wearer from infection is weak, but marginally positive for the use of masks. As a precautionary principle, SAGE, therefore, recommended the use of cloth masks in some higher risk settings where social distancing was not always possible, giving public transport and shops as examples but also advising that any policy decision should not jeopardise the supply of masks to those in settings where there was stronger evidence on their effect e.g. health and care settings [28].

**Table 1. Guidelines and policy changes for face mask use in the UK's devolved nations during 2020.**

| Policy | Devolved nation | | | |
| --- | --- | --- | --- | --- |
| | **England** | **Wales** | **Scotland** | **Northern Ireland** |
| **Face masks recommended in enclosed spaces such as shops, trains and buses** | 11 May 2020 | 9 June 2020 | 28 April 2020 | 7 May 2020 |
| **Face masks mandatory on public transport** (except for those with exceptions) | Announced: 4 June 2020 | Announced: 13 July 2020 | Announced: 18 June 2020 | Announced: 2 July 2020 |
| | Effective from: 15 June 2020 | Effective from: 27 July 2020 | Effective from: 22 June 2020 | Effective from: 10 July 2020 |
| **Face masks mandatory in shops** (except for those with exceptions) | Announced: 14 July 2020 | Announced: 11 September 2020 | Announced: 2 July 2020 | Announced: 6 August 2020 |
| | Effective from: 24 July 2020 | Effective from: 14 September 2020 | Effective from: 10 July 2020 | Effective from: 10 August 2020 |

Data sources: [13–24]

Furthermore, on 4 June 2020, the Environmental Working Group, one of SAGE's sub-committees released a report on COVID-19 transmission, advising that if 2m face-to-face distancing could not be achieved, additional mitigation measures including (but not limited to) face masks and minimising the duration of exposure should be undertaken [29]. At a similar time (5 June 2020), the WHO updated their guidance, which now included advising governments to encourage the general public to wear masks in areas with widespread transmission and where physical distancing was difficult, giving public transport, shops and confined crowded environments as examples. In his opening remarks at a media briefing, the WHO's Director-General stated these changes were based on '*evolving evidence*' [30].

These changes in face mask recommendations for the public required behavioural changes. During (and beyond) our study period, Imperial College London and YouGov's COVID-19 behavioural tracker was a global survey collecting fortnightly data exploring the public's attitudes and health behaviours [31]. When asked '*Thinking about the last 7 days. . .how often have you worn a face mask outside your home to protect yourself or others from coronavirus (COVID-19)?*', in the 15 May 2020 UK survey, 15% answered 'always' and 63% 'not at all'. There was a notable increase in 'always' responses from the 3 July 2020 when 17% answered always, to 55% on the 29 July 2020, with a further 20% answering 'frequently'. These dates correspond with some of the policy changes shown in Table 1.

The Royal Society and British academy [9] reported the uptake of mask use was slow in the UK due to a lack of clear recommendations for the lay public attributed to i) an over reliance on an evidence-based medicine approach and therefore an assertation the evidence was weak; ii) inconsistent and changing advice from supranational organisations, such as the WHO; iii) concern over the applicability of scientific findings in different contexts e.g. countries and types of pandemic; and iv) mix of supply concerns for Personal Protection Equipment (PPE) e.g. difference between surgical face masks for medical professionals and masks recommended to the public. A separate study (conducted 17–20 July 2020) by King's College London and Ipsos MORI [32] showed that 13% of interviewees felt the government only wanted the lay public to wear face masks as a way of control and 10% felt wearing them was bad for their health. Due to these objections and/or hesitancies, mandating the use of face masks did cause controversy for some members of the public [33].

Elsewhere globally, reasons provided for higher levels of adherence of face mask use in China and Hong Kong compared to other countries were government advertising campaigns, health education programs, imposing punishment for those not wearing a mask and previous experiences in outbreak management during the 2003 Severe Acute Respiratory Syndrome (SARS) epidemic [34–37]. In contrast, face masks not being a cultural norm, alongside mask shortages and mixed messaging about their use, were provided as reasons for lower accordance of mask-wearing in comparison to other mitigation strategies in Malaysia [4]. These attitudes and practices towards face mask use are often informed by people's knowledge and risk perceptions [7]. This in turn relates to the public's understanding of science and which can be affected by the information available to them, and therefore the digital information ecology (for those that use the internet).

## Digital information ecology and crisis communication

Following Perreault & Perreault's work on the Communication Ecology of Pandemic reporting [38], we acknowledge that there are several microenvironments within the crisis communication ecology, and this study examines how journalistic outputs, citizen commentary, governmental and public health messaging, as the associated communication practices coincide. Each warrants consideration, as the motivations and agendas of different actors in the network contribute to the

wider communication ecology. Despite increased efforts to understand the digitally mediated communication practices in both scientific and crisis communication contexts, there is no systematised theorisation that attends to these overlapping and related concerns. Developing this understanding is therefore key for effective crisis communication [39].

For this investigation we focus on one platform in this digital information ecology, Twitter. Twitter is a type of microblogging social media platform which is and has been used as a platform for rapid and widespread sharing of information in crisis situations, including previous pandemics such as H1N1/Swine Flu and the Zika outbreak, as well as for controversial health issues such as vaccines [8, 11, 40–42]. Crisis events, such as the COVID-19 pandemic, are often subject to uncertainty, where there is a lack of information and a need for sensemaking [43]. Infodemiology methods can provide '*direct and honest data on health information seeking behaviour*' [44] and be useful for identifying how the public use and share information sources [41]. Using previous Twitter studies as precedents [11, 45–50], we focus on the following in our study: tweet content, frequency, external resources for information, and users circulating information. Each of these aspects of the digital information ecology, alongside relevant theoretical perspectives, are now discussed in turn.

## Tweet content

Separate studies have shown that a high proportion of tweets are informative, meaning they seek to share information or updates [48, 51]. Furthermore, Twitter can be used to address emotional needs e.g. an individual expressing concerns with their online network [43]. Existing theoretical research, on Agenda-Setting, for example, has considered how media reporting, as well as how policy and public agendas, direct public attention towards particular issues. Concluding, topics with high relevance and uncertainty often lead to a higher need for orientation [52]. Identifying the content of Tweets can help to identify topics of interest at particular points in time [53]. For example, one study (conducted 2 February– 15 March 2020) identified twelve topics of concern during the pandemic, one being '*face-mask wearing*'. These '*face-mask wearing*' tweets commonly focused on the importance of mask use in reducing the spread of the virus and shortages in PPE supplies [45].

## Tweet frequency

The ongoing and long-term nature of the COVID-19 pandemic contrasts the majority of existing studies on risk communication which tend to focus on 'short-term' natural hazard events e.g. earthquakes (the event rather than impacts being short-term) [8]. The frequency of tweets about a topic over time can be used as an indication of public attention about that topic. Previous studies have shown that public attention to a topic can be short, referred to as 'attention saturation'. This 'attention' includes the public's awareness, risk perceptions and tendencies towards behavioural changes [54–56]. An example being the heightened attention levels caused by local COVID-19 cases leading to people seeking information about symptoms. As mixed messaging, including inconsistent and unclear messages, from governments has been identified in papers as a hindrance to the public's understanding and adoption of behavioural change [11, 57], including face masks [9], arguments have been put forward encouraging policy makers to take advantage of the period of active information seeking to ensure clear advice is given to the public and/or to reinforce messaging [54, 55].

## External resources of information on Twitter

The rise of the internet and social media over the past few decades has substantially changed the media environment, including the role and input of traditional media (i.e., newspapers

which were traditionally in print). Their role has been challenged by providing other channels to alternative information [58]. This includes the use of external resources and knowledge on Twitter [8]. In crisis situations, this volume of resources is often exacerbated as the public are exposed to more competitive and conflicting information [39]. Concurrently, as outlined in Media Dependence Theory studies, in the absence of credible and complete information from official sources the public seek alternative channels of information [59]. The high volume of information during the COVID-19 pandemic has led to high levels of medical misinformation propagation, with challenges of reducing its spread being coined as an 'infodemic' [47, 51, 60, 61].

One way of capturing the dissemination of external resources and the credibility of information is through the analysis of the links to web addresses, commonly known as URLs. One study of COVID-19 related tweets identified that high quality journals and information were rarely circulated, and identified the top domains mentioned in COVID-19 related tweets were social media websites, including YouTube and Instagram, in addition to broadcasting and news media websites such as the BBC and the Guardian [46]. In another study, links were given scores based on their credibility [47]. They found that links to what they defined as low-credibility sources accounted for 0.89% of the total tweet volume, which was comparable to the volume share (0.98%) of the New York Times, a traditional media outlet which they describe as a reliable source. They also found low-credibility sources to have a higher tweet volume than the Centre for Disease Control (CDC), the USA's health protection agency, which accounted for 0.65%. Another study expresses a preference towards sharing tweets containing official government-based resources and evidence-based public health information after their analysis of tweets by G7 world leaders, in which only 28.6% of tweets classified as informative contained links to official sources [48].

## Active participants in information circulation

The Social-Mediated Crisis Communication (SMCC) model conceptualises 'information source' as where the crisis information originates. This can be either an organisation experiencing the crisis or a third party, including a journalist or social media creator [62]. Alongside the diversity of digital platforms, changing models of scientific crisis communication and intensifying digitalisation have increased the diversity of communicators and these 'information sources'. It is now possible for anyone to enter the '*communications fray*' and '*audiences are turning into active participants*' in public discourses on science–this has led to a pluralistic and participatory digital media ecology for science [63], but also increasing polarising views and controversy [38].

A critique of crisis communication theory is the focus on organisations, rather than how publics communicate with each other, leading to calls to focus on multiple actors engaged in crisis response [62, 64]. Furthermore, within this changing digital information system, Agenda-Setting theory recognises that these different actors (i.e., news media and individuals) are unlikely to have the same agendas when disseminating information [43]. Within this theoretical framework, there have been suggestions that traditional news media have lost some of their power and visibility in information circulation due to the plurality of voices on Twitter. However, some studies, foci including COVID-19 and climate change, have found that traditional news media still have a high profile–thus the increase of 'information sources' and actors has modified the traditional media's role rather than eliminated it. [43, 63, 65]. Within this information landscape, concerns have also been expressed that science communicators are not achieving a high profile due to the crowded digital ecology, yet there are arguments that there is scope for them to do so [63].

There is commonly a perception that official sources are more credible than unofficial sources and that government health agency and news media are more successful than their social peers for 'debunking misinformation' in a public health crisis [39]. In the context of COVID-19, one study (conducted 27 February 2020 using a sample of 673 tweets) [51] found a high prevalence of informative tweets were from 'informal individuals or groups' (66.6%) and 'news outlets or journalists' (16.5%). In contrast to this study, an analysis of tweets collected from 21 January—31 March 2020 found the most active users were from verified accounts belonging to news sources and political figures, commenting that these influential figures and platforms often use Twitter to disseminate breaking news and amplify their messaging [49]. However, even if verified, studies have also shown that tweets from political figures can be prone to misinformation, which can easily be amplified [50]. Due to this potential influence, recommendations have been made to closely monitor branded organisations and celebrity user accounts to prevent the spread of misinformation at an early stage [61].

Our own research uses the concept of the digital information ecology and these infodemiology methods to frame, and inform, our analysis of Twitter data focusing on face mask specific tweets. We evaluate what got tweeted and by whom, aiming to develop understanding on how these information sharing practices could have affected the public's understanding of the scientific evidence and expertise in relation to face masks. The following section outlines the methods used to answer the research questions outlined in the introduction.

## Method

### Data collection and 'Twitter datasets'

Our analysis utilises Banda et al.'s [66] COVID-19 Twitter chatter dataset which includes IDs of tweets relating to COVID-19. We used a subset of these tweet IDs to create a dataset of tweets covering the period between 22 January and 1 August 2020. Data was collected via the Twitter Application Programming Interface (API) using the python library Twarc, which allows you to search for tweets by ID and retrieve the tweet text and associated metadata, which included the date the tweet was created at, the number of retweets it received, and the user profile description.

To focus on discussion relating to the use of face masks, the dataset was filtered based on a selection of keywords. Specifically, only tweets that mention 'face masks' or 'face coverings', and possible spelling variations on these keywords (e.g. facemask, face-masks, face-covering, face-coverings, etc. . .) were included. To focus the dataset more closely on the UK-specific discussion of face masks, the data was filtered in several ways. Where available, geotag data or the optional location field in user profiles was used. While location data from profiles is self-reported by the user and may not always be accurate or up to date, it provides some indicator of location or affinity with the UK. Additionally, the dataset was filtered to only include English language tweets. Retweets and duplicates were also removed. This resulted in a total dataset of 95,876 unique tweets, which we call the 'face mask tweets' dataset.

In total, there are four 'Twitter datasets' referred to in the analysis section: 'face mask tweets', 'most retweeted tweets', 'most tweeted URLs' and 'scientific evidence and expertise URL titles'. As described above, the 'face mask tweets' dataset includes all twitter data collected and filtered to face mask specific tweets from the UK. As the main interest of this study is the circulation of information about face masks the remaining three datasets focus exclusively on tweets that contain an external URL. External URLs included in tweets provide links to information outside of Twitter itself and are now a common part of Twitter practice, particularly when a user finds a topic newsworthy [67]. In total, 45,466 tweets containing an external URL were identified and 41,629 unique URLs were embedded within these.

The 'face mask tweets' dataset was first filtered to remove any tweets which did not include an external URL (i.e. removing those with no URL or a link to another tweet rather than source outside of Twitter), before the two separate sample datasets were created. Samples were then taken to facilitate a manual analysis. The 'most retweeted tweets' dataset sample contains tweets with an external URL which have more than ten retweets, accounting for 2,333 tweets, and representing 5.13% of the total 'face mask tweets' containing a URL. The 'most tweeted URLs' are URLS contained within more than 10 unique tweets in the 'face mask tweets' dataset. This dataset contains 333 URLs, accounting for 0.73% of the unique URLs. Previous studies have shown, the use of the retweet count (used in the 'most retweeted tweets' dataset) can be an indication of tweets that the most users express: the significance of information to their followers, feelings and/or get feedback/alert others [68, 69]. Retweets are used rather than likes in this paper as the focus is on the circulation of information, hence they are an important aspect of the social media information ecology [11, 42]. The 'most tweeted URLs' originate from users selecting links for inclusion and deeming them newsworthy enough to share with their network. The URL count is an indication of the number of tweets where different Twitter users chose to include a specific URL in their own tweet.

In both the 'most retweeted tweets' and 'most tweeted URLs' datasets the URL subject content was characterised (process described in the next section). To focus on 'scientific evidence and expertise', both datasets were filtered further to those where the URL subject matter was categorised as this i.e., reports on/discusses scientific evidence and/or expertise. As some URLs led to the same website, one reason being shortened URLs, duplicate titles were removed and for each title, total retweets and number of unique tweets containing the URL were calculated. This process of removing duplicates and identifying the shortened URLs was not undertaken for the other datasets due to feasibility of manually checking each title was unique and ensuring different links were not accidentally merged. In total 230 URL titles were identified across the 'most retweeted tweets' and 'most tweeted URLs' datasets categorised as scientific evidence and/or expertise. The datasets outlined were analysed in a variety of ways, which are now described.

## Data analysis

An overview of the content analysed in each of the datasets described is provided in Table 2. This includes: topics and keywords within tweets, URL subject content, views towards face mask use for those categorised as 'scientific evidence and expertise', overall frequency of tweets, URL domains, and user profiles sharing URLs. Excluding the full 'face mask tweets'

**Table 2. Twitter datasets and content analysed.**

| Dataset | Description | Number of data points | Content analysed |
|---|---|---|---|
| Face mask tweets | All tweets containing keywords e.g. mask and covering. | 95,876 tweets 45,466 containing a URL 41,629 unique URLs | Topic modelling Frequency of tweets |
| 'Most retweeted tweets' sample | Tweets from 'face mask tweets' dataset containing an external URL and with more than 10 retweets | 2,333 tweets containing a URL (5.13% of tweets containing a URL) | URL subject content Domains User profiles |
| 'Most tweeted URLs' sample | External URLs contained within more than 10 unique tweets in the 'face mask tweets' dataset. | 333 URLs (0.73% of unique URLS) | URL subject content Domains |
| 'Scientific evidence and expertise URL titles' sample | URL titles from the 'most retweeted tweets' and 'most tweeted URLs' categorised as 'scientific evidence and/or expertise' | 230 unique URL titles | Domains User profiles ('most retweeted tweets' only) Views towards face mask use |

**Table 3. Timeframes used for analysis.**

| Timeframe (TF) | Dates | Reasons |
|---|---|---|
| TF1 | 22 January– 27th April 2020 | The period of time before the first announcement was made by one of the devolved nations, Scotland on 28th April 2020, that face masks were recommended on public transport. |
| TF2 | 28th April– 3rd June 2020. | The period of time from the first announcement recommending face mask use to the announcement that face masks would be mandatory on public transport by one of the devolved nations, made by England on the 4th June 2020. |
| TF3 | 4th June– 1st August 2020 | The period of time after the first announcement face masks would be mandatory on public transport by one of the devolved nations to the end-date of data collection, 1 August 2020. When data collection ceased, face masks were mandatory on public transport in all the devolved nations and were mandatory in shops in England, Scotland and Wales. |

and user profile names (see sections about ethical and Twitter data considerations), all of these datasets are provided in S1 Appendix.

In this study, three timeframes (TFs) are used to separate the data (Table 3), corresponding with the first announcements of changes to the face mask recommendations and policy by one of the UK's devolved nations.

**Pre-processing and topic modelling.** The 'face mask tweets' dataset was pre-processed to remove URLs and stop words (i.e. words that feature commonly but do not include semantically rich information), all text was converted to lower case and the data were tokenised and lemmatised, grouping words that have the same stem (i.e. 'thanking', 'thanks' become 'thank'). Due to the size of the 'face mask tweets' dataset, computational approaches for text analysis were employed. To identify common topics discussed within the data and help contextualise our study, we applied a commonly used probabilistic topic modelling for text analysis [53]. Latent Dirichlet Allocation (LDA)-based topic modelling is an unsupervised machine learning technique for the automatic description of documents [70]. Applying LDA to the dataset of tweets returns a set of topics, with each tweet described as a distribution of these topics. A label is generated by the researchers after manually inspecting 50 examples with the highest topic contribution for each topic. Each topic is also assigned a weighting to reflect how prominent it is in the observed data instance.

The LDA model was implemented using the python package Gensim [71] (with parameters: n-grams 1–3, passes = 20, iterations = 100, random_state = 100). The number of topics was manually selected based on interpretability and performance, with 10 topics providing the most coherent and interpretable output.

**URL subject content.** The URL subject indicates the content of the information to outside resources being shared and/or circulated by Twitter users. For both the 'most retweeted tweets' and 'most tweeted URLs' datasets, the URLs titles were manually collected. A manual inductive coding process was then used to categorise the content of URLs based on their heading and sub-headings. Rather than using a pre-defined list, 'researcher 1' analysed all the URLs and identified emerging themes, whilst 'researcher 2' coded a random sub-sample (approximately 10%) of the data. After this first step, the categories were then refined following discussion between the researchers. Using the refined list of codes, the process of categorising the URLs was repeated. A full list of these codes and descriptions are provided in S2 Appendix.

To validate the coding between the two researchers, their categorisations were compared (see S3 Appendix). For the URL subject content's main themes, our agreement was 75.91%, whilst sub-categories (of which there were more choices) was 64.55%. These variations were

then discussed by the two researchers to identify if there was significant disagreement or if different categorisations were due to multiple categories being applicable and the inherent subjective nature of inductive coding. The latter was always the case.

**Scientific evidence and expertise.** To facilitate a greater understanding of the scientific information being circulated, the top 25 URL titles in the 'most retweeted tweets' and 'most tweeted URLs' datasets (ranked by number of retweets or tweets) focusing on 'scientific evidence and expertise' were examined. By reading the entirety of their content we identified if the article supported/recommended the use of masks for the lay public, did not recommend/support use, or stated the evidence was inconclusive. The top 25 were chosen as they provided what we deemed a sufficient overview of the most circulated URLs, yet was feasible to read through the entirety of the content.

**Frequency of tweets.** The overall frequency of tweets indicates the interest in face masks (determined by the presence of face mask/covering keywords in the text). Generating a frequency count of the total number of tweets per day in the 'face mask tweets' dataset provides an overview of changes in volume of tweet activity relating to face masks in the period studied (22 January—1 August 2020), and whether these corresponded with key guidance/policy changes. We also include Google Trends [72] data as an indicator of people's search interest in 'face masks' or 'face coverings' on Google's search engine.

Furthermore, due to the evolving nature of the scientific evidence and change in guidance, an analysis was undertaken examining the point in time the links referring to scientific evidence and/or expertise were shared. Using the date when the associated tweets were created at, it was possible to analyse when URLs appeared over the study time period.

**Domains.** Categorising URL domains explores what type of content and information resources are being retweeted or shared [46–48]. We obtained descriptions of the domains from either the website itself or alternative sources (also available in S1 Appendix). Using the same manual coding process as the URL subject content, the domains in the 'most retweeted tweets' and 'most tweeted URLs' were then classified based on the researchers' interpretation of descriptions of the domain host available on the domain website (see S2 Appendix for full list of categories). Our inter-annotator agreement was 76.79%. As with the subject categories, discrepancies were due to the applicability of several categories to one domain rather than a significant disagreement.

**User profiles.** The analysis of user profiles allows for a better understanding of who is circulating information [61] and who the 'major players' are [42]. These were also manually examined using an inductive coding technique based on the user profile description at the time the dataset was created (August 2020). Each user was identified as either an individual or an organisation, and then the professional background assigned (see S2 Appendix). Unlike, URL subject content and domains, this analysis was limited to the 'most retweeted tweets' dataset. The inter-annotator agreement was 77.11% for the profession categories, which again were corroborated to ensure there were no significant disagreements.

## Ethical considerations

Ethics approval was obtained by the lead researchers' department, the Centre for Research in the Arts, Social Sciences and Humanities (CRASSH) at the University of Cambridge. As part of these considerations, informed consent was not deemed necessary as full tweet content has not been shared, nor the usernames associated with Twitter accounts of individual people (rather than organisations) in the paper or within the Supporting Information. Those individual accounts that may be identifiable from the discussion are those with a large following or interaction rate. The data has been stored on a password protected computer and cloud server. The ethics statement was approved on these principles.

## Twitter data considerations

The study uses publicly available data in accordance with the Content Redistribution clause under Twitter's Developer Agreement and Policy [73]. In compliance with Twitter's Terms and Conditions, we are unable to publicly release the text of the collected tweets. However our publicly available dataset (S1 Appendix–Excel Tab 'Most Retweeted Tweet dataset', accessible via PLOS ONE's server) includes the Tweet IDs, which are unique identifiers tied to specific tweets, of publicly available posts, in compliance with the Twitter Terms of Service [73]. These are openly available for others to use in compliance with Twitter's Terms and Conditions. As previously outlined, our analysis used Banda et al.'s [66] COVID-19 Twitter chatter dataset as a starting point.

## Limitations

While studies of social media can provide insights into public opinion and discussion on a given topic there are limitations. Firstly, the keyword sampling and data filtering techniques, such as excluding any tweet that did not provide some indication that the author was associated with the UK, will result in a bias in the type of data collected. Consequently, it is inevitable that such an approach will only capture a proportion of the discussion. Secondly, while the study can provide insight into information sharing practices on Twitter, the information ecology is clearly much more complex and comprised of multiple different social media and messaging platforms through which information is exchanged. Thirdly, our study is limited to the study period (22 January 2020–1 August 2020). As social distancing restrictions eased over the summer of 2021, including the change back from mandatory use to recommending use of face masks in indoor spaces in England, there was a re-emergence of tweets about face masks, as well as accusations in the media about mixed messaging [74]. Lastly, the manual analysis focuses on relatively small samples of data. This enabled a more qualitative exploration of the content than could otherwise have been achieved through automated techniques. As this takes an inductive approach for the interpretation and classification of content, there is always an element of subjectivity in the process. For instance, other researchers may have categorised the content using different terminology [48]. The subjective nature of classifying content can also be seen via differentiations in our own inter-annotator agreement scores (S3 Appendix). Despite these limitations, the universal character of our study contributes to developing understanding of the digital information ecology and provides practical insights for crisis communicators.

## Analysis of Twitter content

### Topic modelling and URL subject content

Topic modelling was used to identify common topics discussed within the data which helps to provide an overview on different aspects associated with face masks being discussed. The results of the ten topic LDA topic model, with the salient keywords for each of the topics are displayed in Table 4 (further data provided in S4 Appendix).

Overtime, we found there were downward trends in topics 6, 9 and 10 which relate to advocating for the use of face masks, PPE supplies and the global situation including the number of COVID-19 cases around the world. Upward trends are observed in topics 2, 5 and 7, categorised as: the adoption of mask wearing and a mix of discussion of pros and cons; shopping and resuming everyday activities with face masks; and announcements about the mandatory use of masks.

**Table 4. Topic modelling categories.**

|  | Topic 1 | Topic 2 | Topic 3 | Topic 4 | Topic 5 |
|---|---|---|---|---|---|
| Keywords | People, get, would, need, virus, stop, protect, think, know, go | people, say, government, could, take, see, get, right, make, would | trump, protective, american, say, refuse, president, america, pandemic, call, season | wearamask, man, pandemic walk, fit, stayhome video, street, ventilator, watch | go, lockdown back work, get, time, day, shop, rule, come |
| Label | Wear a mask, anger at others not wearing mask | Mixed discussion on mask adoption, citing pros and cons | American mask policy and Donald Trump mask use | Politics, police, racial inequality | Shopping and resuming normal practices with facemasks |
|  | Topic 6 | Topic 7 | Topic 8 | Topic 9 | Topic 10 |
| Keywords | staff, keep, safe, home, nhs, stay, hospital, worker, protect, glove | public, spread, transport, mandatory, use, make, say, reduce, compulsory, help | bbc_new, police, london, nhs_trust_boss, meet, useless, fail, maskup, word | ppe, make, nhs, use, supply, help, protection, support, buy, surgical. | Case, country, china, death, new, state, world, report, crisis, outbreak |
| Label | Encouraging face mask use and other mitigation strategies inc. PPE | Mandatory use of face mask announcements | Types of masks available, mask fashion and making | Mask sales, PPE and NHS | Global situation, political figures |

A variety of themes were also identified from the inductive analysis of URL subject content, with some themes comparable to the topic modelling e.g. downward trend for PPE supplies, and upward trend for guidance and policy related content (see Fig 1, with further data provided in S4 Appendix).

After the first announcement of masks being mandated (the third timeframe), the topics became dominated by information related to guidance and policy, scientific evidence and expertise, and COVID-19 cases and guidelines in other countries. Art is also dominant here due to the most retweeted tweet (Table 5) containing a URL about a well-known artist, Banksy: '*Coronavirus*: *New Banksy piece on London Tube encourages people to wear masks*' [75]. The tweet received 10,786 retweets, with the next most retweeted tweet receiving 3,194 retweets. Meanwhile, other topics including 'exemptions and disability concerns' have not been discussed as much in comparison.

There is a notable difference in the proportions of retweets/tweets relating to 'guidance and policy' when comparing the overarching themes for the 'most retweeted tweets' and 'most

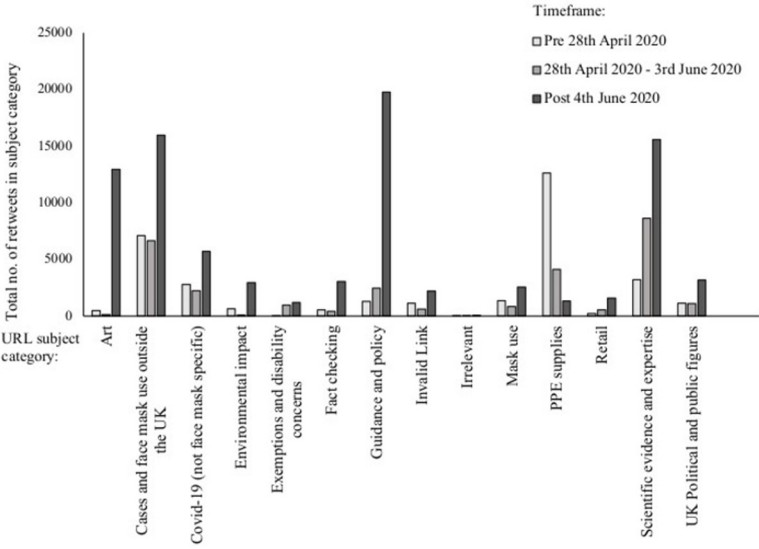

**Fig 1. Subject categories in the 'most retweeted tweets' dataset.**

**Table 5. URL titles of the top 5 'most retweeted tweets' (subheadings excluded).**

| Tweet ID | Tweet date | URL Title | Domain* | Date of publication | Theme | Retweet Count |
|---|---|---|---|---|---|---|
| 31747 | 15/07/2020 | **Coronavirus: New Banksy piece on London Tube encourages people to wear masks [75]** | https://www.news.sky.com | 15/07/2020 | Art | 10786 |
| 55855 | 01/06/2020 | **Physical distancing, face masks, and eye protection to prevent person-to-person transmission of SARS-CoV-2 and COVID-19: a systematic review and meta-analysis [76]** | https://www.thelancet.com | 01/06/2020 | Scientific evidence and expertise | 3194 |
| 55850 | 22/04/2020 | **America's Heroism Trap [77]** | https://slate.com | 20/04/2020 | Cases and face mask use outside the UK | 3090 |
| 55863 | 27/05/2020 | **Reducing transmission of SARS-CoV-2 [78]** | https://www.science.sciencemag.org | 26/06/2020 | Scientific evidence and expertise | 2036 |
| 19302 | 08/07/2020 | **Coronavirus: 'The masks you throw away could end up killing a whale' [79]** | https://bbc.in | 07/07/2020 | Environmental impact | 1956 |

*Domain of original article rather than shortened URL

tweeted URL' datasets (see S4 Appendix), with a higher proportion of 'most tweeted URLs' being categorised as 'guidance and policy' respective to the other categories in the dataset. As with the 'most retweeted tweets', the subject content categorised as 'scientific evidence and expertise' was popular in comparison to some of the other categories in the 'most tweeted URLs' dataset. Overall, 18.36% of the 'most retweeted tweets' and 12.87% of the 'most tweeted URLs' were categorised as 'scientific evidence and expertise'.

Clearly, several topics related to face masks were discussed and circulated on Twitter with the communication of science becoming particularly evident as the guidelines changed. From those ranked in the top 25 according to retweet count categorised as 'scientific evidence and expertise', 15 supported use for the general public, 8 did not, 1 said the evidence was inconclusive, and another focused solely on the use of masks for medical staff. For the 'most tweeted URLs', 19 supported use, 4 did not support and two said the evidence was inconclusive. This variation of recommending/not recommending (or supporting/not supporting) use is also reflecting in Tables 6 and 7 which show the top five URL titles based on the sum of retweets and number of unique tweets containing the URL (all 25 URL titles are provided in S1 Appendix).

By focusing on the external resources categorised as 'scientific evidence and expertise', it is clear there was mixed messaging about their use. As the guidance changed overtime, we now

**Table 6. Most retweeted URL titles (duplicates combined) for URLs classified as 'scientific evidence and expertise' in 'most retweeted tweets' dataset.**

| Title | Domain | Date published | Sum of retweets | Sub-theme | View towards masks |
|---|---|---|---|---|---|
| Physical distancing, face masks, and eye protection to prevent person-to-person transmission of SARS-CoV-2 and COVID-19: a systematic review and meta-analysis [76] | https://www.thelancet.com | 01/06/2020 | 3338 | Scientific studies | **Supports use** |
| Reducing transmission of SARS-CoV-2 [78] | https://science.sciencemag.org | 26/06/2020 | 2688 | Scientific studies | **Supports use** |
| Coronavirus: 'We do not recommend face masks for general wearing' [80] | https://www.bbc.co.uk | 03/04/2020 | 1852 | Expert opinion | **Not supported** |
| Coronavirus: Face masks could increase risk of infection, medical chief warns [81] | https://www.independent.co.uk | 12/03/2020 | 1020 | Expert opinion | **Not supported** |
| How the World Missed COVID-19's Silent Spread [82] | https://www.nytimes.com | 27/06/2020 | 911 | Expert opinion | **Supports use** |

**Table 7. URL titles (duplicates combined) contained in most tweets for URLs classified as 'scientific evidence and expertise' in 'most tweeted URLs' dataset.**

| Title | Domain | Date published | Total number of tweets | Sub-theme | View towards masks |
|---|---|---|---|---|---|
| Coronavirus: Face masks could increase risk of infection, medical chief warns [81] | https://www.independent.co.uk | 12/03/2020 | 135 | Expert opinion | **Not supported** |
| Coronavirus: Wear masks in crowded public spaces, says science body [83] | https://www.bbc.co.uk | 07/07/2020 | 117 | Expert opinion | **Supports use** |
| Coronavirus: Wearing surgical masks can reduce COVID-19 spread by 75%, study claims [84] | https://news.sky.com | 19/05/2020 | 63 | Scientific studies | **Supports use** |
| Oxford COVID-19 study: face masks and coverings work–act now [85] | https://www.ox.ac.uk | 08/07/2020 | 53 | Scientific studies | **Supports use** |
| To help stop coronavirus, everyone should be wearing face masks. The science is clear [86] | https://www.theguardian.com | 04/04/2020 | 50 | Scientific studies | **Supports use** |

focus on when information was being circulated and the public were seeking/sharing information.

## Frequency of tweets and circulation of scientific resources

Face mask specific tweet frequency indicates the interest in face masks overtime. The frequency of tweets with and without a URL within the 'face mask tweets' dataset is presented in Fig 2A, alongside annotations of the key guidance and policy changes in England (raw data is provided in S4 Appendix). Fig 2B. includes Google Trend [72] data which indicates the public's search engine behaviour/interest in the topics 'face masks' and 'face coverings'.

For both tweet frequency and Google Trends [72] data, the results show clear peaks and troughs in either frequency or search interest corresponding with England's guidance/policy changes, with the biggest peak for tweet frequency being 14 July 2020 when it was announced that face masks would become mandatory in England's shops. Fig 2A also shows the proportion of tweets containing a URL. Overall, 46.58% of the tweets contained a URL but this decreased across the three timeframes posed (pre and post recommendations and mandating mask use).

When focusing on the external resources categorised as 'scientific evidence and expertise', we identified that some of the URLs, particularly those published earlier in the pandemic contained an editor's note recognising that the scientific evidence had developed since the publication of the article. For example, in the URL entitled 'Coronavirus: 'We do not recommend face masks for general wearing', it now states 'Update: This video contains advice that has since been updated—people across the UK now have to wear face coverings in certain circumstances when out of the house' [80]. For this reason, we analysed the point in time that the URLs were shared. Fig 3 shows the results for two of the articles not recommending use [80, 81].

In the case of the URLs shown in Fig 3, they resurfaced on Twitter after their initial publication, this was when the policy changed (or policy changes were announced) and face masks became mandatory in shops. The external URLs were contained in more tweets at this later point in time than their original publication date. A review of these tweets indicate that people were mainly sharing the articles alongside statements claiming that face masks were not beneficial and using them as evidence in arguments with other users, often referencing the expert status of the source (e.g. 'England's most senior doctor'). A smaller proportion referenced the articles in tweets calling attention to and questioning the changing guidance and recommendations.

These results provide useful insight into the ways public interest intensifies at certain points in a crisis, in addition to the inclusion of external URLs as a key information sharing practice.

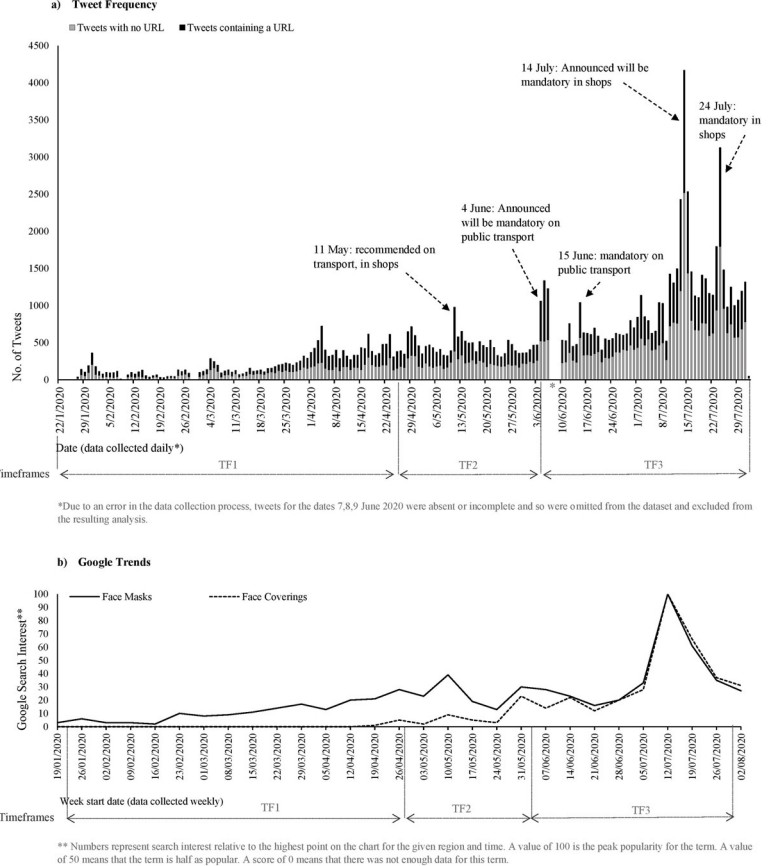

**Fig 2.** a) Frequency of 'face mask tweets' with and without URLs. Annotated with key guidance and policy changes in England b) Google Trends data for search terms 'face masks' and 'face coverings' in the UK (Data source: Google Trends [72]).

We have also identified that challenges can arise due to the prolonged circulation of out-of-date information. We now move on to identify the sources of these external resources feeding into Twitter discussions.

## Popular domains

Beyond the content of external URLs, it is also important to understand the creators of the information resources being retweeted. We use domains as an indication of this. Fig 4 (with further detail provided in S4 Appendix) provides a comparison between the percentage of retweets/tweets in the domain categories in the 'most retweeted tweets' and 'most tweeted URLs' datasets.

Many domains identified in the analysis belong to journalism or broadcasting websites. In the 'most tweeted URLs' dataset, there is more of a spread of domain categories, although journalism and broadcasting remain dominant. Excluding URL shortening, official government websites were the next most retweeted/tweeted domain category but noticeably lower. For the filtered scientific evidence and expertise dataset, the proportion of domains for academic, scientific journals and organisations is greater, but journalism and broadcasting both remain dominant.

To provide more information on what the most popular domains were, the top five domains based on the total number of retweets/tweets for the overall datasets and those classified as 'scientific evidence and expertise' within these are shown in Table 8.

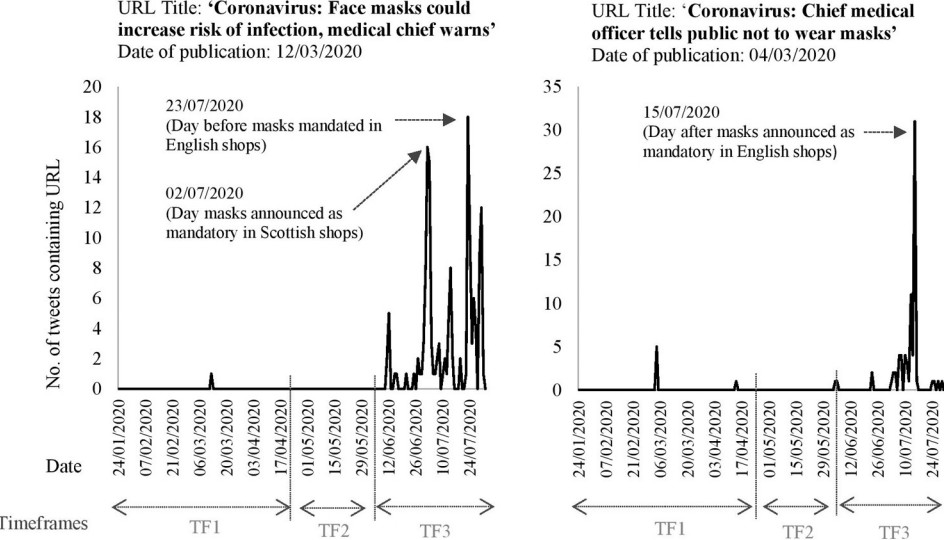

**Fig 3. Recirculation of two URLs about scientific experts not recommending face mask use.** Annotated with corresponding policy/guidance changes.

Within the highest ranked domains, those categorised as journalism or broadcasting include: The Independent, The Guardian, The Daily Mirror and the BBC. In a survey (conducted April 2019 –March 2020) assessing the monthly reach of leading national newspapers in the United Kingdom [87], The Independent, Guardian and The Daily Mirror are ranked within the top 5 (taking into account phones, tablets, desktops and prints). However, the two highest ranking newspapers in the survey, The Daily Mail and The Sun, only account for

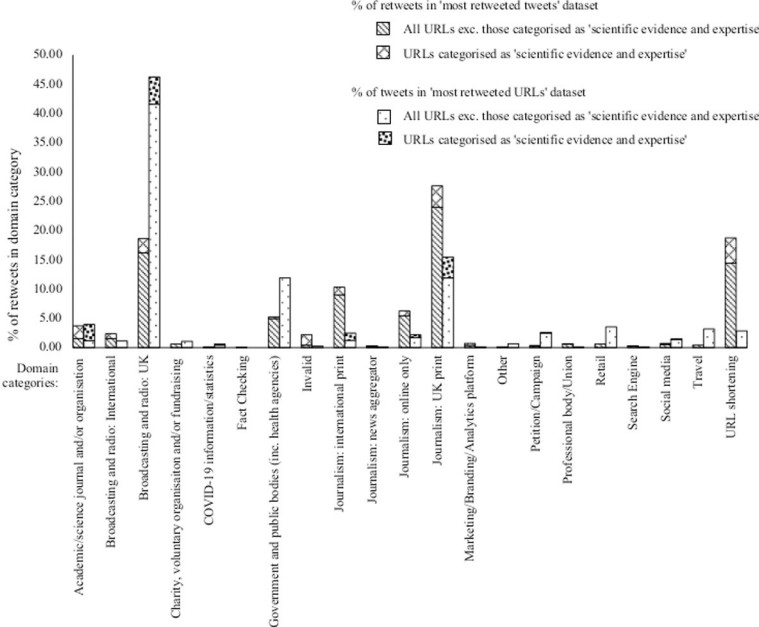

**Fig 4. Comparison between the percentage of retweets/tweets in the domain categories in the 'most retweeted tweets' and 'most tweeted URLs' datasets and identifying whether the URL subject content was categorised as 'scientific evidence and expertise'.**

**Table 8. Top five domains in each dataset based on total number of retweets/tweets within dataset.**

| All ('URL shortening' domains and duplicates not removed) | | | | | |
|---|---|---|---|---|---|
| Most retweeted tweets | | | Most tweeted URLs | | |
| Domain | Domain category | Proportion of total retweets in dataset (shown as % of dataset) n = 139,000 retweets | Domain | Domain category | Proportion of total tweets in dataset (shown as % of dataset) n = 9,502 tweets |
| https://www.independent.co.uk | Journalism: UK print | 10.15 | https://www.bbc.co.uk | Broadcasting and radio: UK | 35.86 |
| https://bbc.in | Broadcasting and radio: UK | 8.11 | https://www.theguardian.com | Journalism: UK print | 9.26 |
| https://www.theguardian.com | Journalism: UK print | 6.43 | https://www.independent.co.uk | Journalism: UK print | 4.78 |
| https://trib.al | URL shortening | 6.18 | http://news.sky.com | Broadcasting and radio: UK | 4.68 |
| https://www.mirror.co.uk | Broadcasting: UK | 3.23 | https://news.sky.com | Broadcasting and radio: UK | 3.16 |
| Filtered scientific evidence and expertise URL titles ('URL shortening' domains and duplicate domains removed) | | | | | |
| | | n = 23,553 retweets | | | n = 1189 tweets |
| https://www.thelancet.com | Academic/science journal and/or organisation | 15.68 | https://www.bbc.co.uk | Broadcasting and radio: UK | 17.49 |
| https://www.independent.co.uk | Journalism: UK print | 9.37 | https://www.independent.co.uk | Journalism: UK print | 13.12 |
| https://science.sciencemag.org | Journalism: Academic/ science journal and/or organisation | 8.75 | https://news.sky.com | Broadcasting and radio: UK | 10.60 |
| https://www.theguardian.com | Journalism: UK print | 4.99 | https://www.theguardian.com | Journalism: UK print | 8.16 |
| https://www.nytimes.com | Journalism: International print | 4.01 | https://www.ox.ac.uk | Academic/science journal and/or organisation | 4.46 |

1.17% and 0.96% of total retweets in our 'most retweeted tweets' dataset (ranks 20[th] and 24[th]). Therefore, these results suggest that the reach of these newspapers may vary depending on the type of platform being used or topics being discussed.

Turning to official resources for information, the WHO's (https://www.who.int) and UK Government's official domains (https://www.gov.uk) rank 34[th] and 14[th] in the 'most retweeted tweets' and 6[th] and 7[th] in the 'most tweeted URLs'. When filtered to scientific evidence and expertise (and shortened URLs had been identified), the Lancet, an academic journal, ranks top of the 'most retweeted tweets'. Oxford University was 5[th] in the 'most tweeted URLs'. The WHO ranks at 17[th] in the 'most retweeted tweets' but does not feature in the 'most tweeted URL' sample, and the UK government domain does not feature in either filtered 'scientific evidence and expertise' dataset samples. We now discuss the users that are sharing this information.

## User profiles with the most interaction

The analysis of user profiles indicates the type of people and accounts that chose to tweet about face masks and who were subsequently retweeted. The distribution of retweets attached to organisations compared to individuals was 58.16% and 41.84% respectively. When this was filtered to those categorised as 'scientific evidence and expertise' the distribution was similar: 54.24% and 45.76%. There was also variation in the user profession categories (see Fig 5 with further detail provided in S4 Appendix).

The majority (73.87%) of user profiles classified as organisations in the 'most retweeted tweets' dataset represents broadcasting/journalism platforms, whilst 11.49% represents

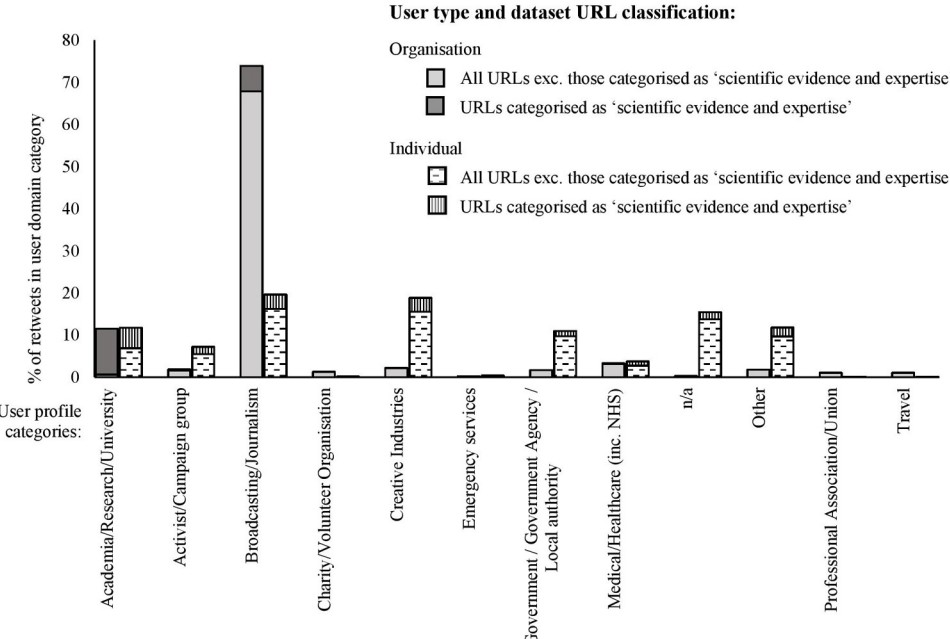

**Fig 5. User profile profession categories and percentage of retweets for those classified as 'organisations' and 'individuals'.** User accounts classified as n/a are not included.

academic institutions. At an individual level, the types of users were more evenly spread between the categories. Notable differences include a higher proportion of those in the creative industries (meaning those describing themselves as an artist, author or similar profession); political/government positions; and medical and health care professionals. The proportion representing academia/research is similar to the organisational level. Due to these differences, the proportion representing the broadcasting/journalism profession is lower for those accounts belonging to individuals compared to organisations.

When filtered to focus on those URLs classified as 'scientific evidence and expertise' there is a large proportion (63.11%) of retweets, particularly at the organisational level, for those representing academia, research or universities in comparison to 34.91% from broadcasting and journalism organisations. At the individual level this breakdown is 24.91% and 17.33%. Although individuals connected to government/local authorities represent 6.30% of total retweets of URLs about 'scientific evidence and expertise', government organisations were at 0.0% in the sample taken.

As with the domains, it is useful to see examples of users in these categories. The top five user profiles ranked by total number of retweets (Fig 6A) are all organisations representing broadcasting or journalism platforms: SkyNews, BBCWorld, Independent, Guardian and BBCWorld. When extended to the Top 10 ranking user profiles, this list includes The Lancet (an academic journal) and individual user accounts. Based on their user profile descriptions, these individuals include an author & public speaker, whose Twitter account has since been suspended; a university professor; a former medic; and a journalist. If focusing on those categorised as 'scientific evidence and expertise' (Fig 6B) there are more accounts representing academia/research/universities and an additional account for an individual person within the top ranking 5. In the top 10, there is another individual account, which has now been suspended from Twitter. The dominance of a small group of user accounts over the total retweet volume can also be seen in Fig 6A and 6B.

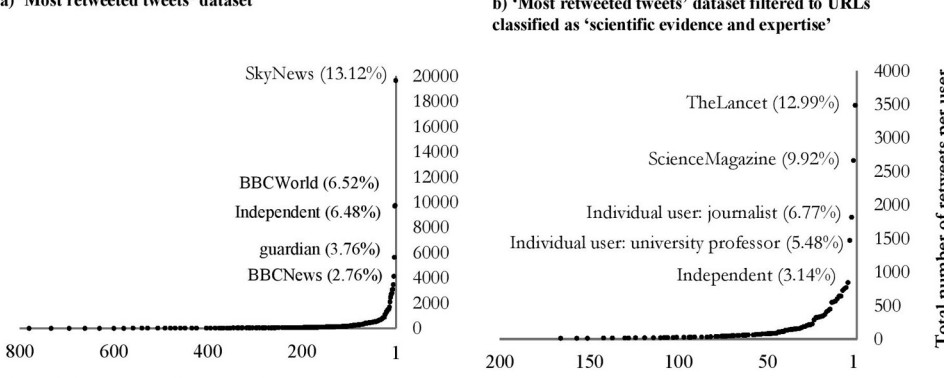

**Fig 6. User profile ranking by number of retweets (x axis) and total number of retweets (y-axis).** a) 'Most retweeted tweets' dataset b) 'Most retweeted tweets' dataset filtered to URLs classified as 'scientific evidence and expertise'.

This analysis section of the paper has presented our results for a range of aspects associated with the circulation of tweets and external URLs focusing on face masks during the COVID-19 pandemic. We now use this analysis to answer the research questions posed at the beginning of this paper.

## Discussion

An overview of the research questions, datasets, methods used and key findings from our analysis is provided in Table 9. We discuss each of these in turn, including the theoretical and practical implications, in the following sub-sections. As a result of our findings, we also pose new

**Table 9. Overview of research questions, applicable datasets, content analysed & methods and key findings.**

| Research question | Applicable datasets | Content analysed & methods | Key findings |
|---|---|---|---|
| What type of information about face masks was deemed as newsworthy and shareable? | Face mask tweets<br>'Most retweeted tweets' sample<br>'Scientific evidence and expertise URL titles' sample | Key word analysis and Latent Dirichlet Allocation (LDA)-based topic modelling.<br>URL subject content (manual inductive analysis)<br>Views towards facemasks e.g. recommending or not recommending (manual analysis) | Range of topics in context of face masks identified. These changed over duration of pandemic e.g. downward trend PPE discussions, upward trend guidance and policy and scientific evidence/expertise.<br>Change in scientific evidence and guidance clear as mix of articles recommending/not recommending face mask use were circulated. |
| When did users choose to share information about face masks during the COVID-19 pandemic? | Face mask tweets<br>Google Trends data<br>'Scientific evidence and expertise URL titles' sample | Frequency count generated showing total number of tweets per day.<br>Weekly search interest and week start date.<br>Point in time Top 25 ranking URLs categorised as 'scientific evidence and expertise' were shared. | Peaks and troughs in the number of tweets and relative search interest corresponded with face mask guidance and policy changes.<br>Circulation of out-of-date scientific information occurred when guidance and policy changed. |
| What are the main external resources of information about face masks? | 'Most retweeted tweets' sample<br>'Most tweeted URLs' sample<br>'Scientific evidence and expertise URL titles' sample | Domains (manual inductive analysis) | Range of external resources were circulated. Dominated by external links to traditional media, but examples of scientific journals/journalism domains and an individual's website (i.e., blog) were widely circulated alongside these. |
| Which user's tweets were most widely circulated? | 'Most retweeted tweets' sample<br>'Scientific evidence and expertise URL titles' sample ('most retweeted tweets' only) | User profiles (manual inductive analysis) | User accounts associated with traditional media, but also scientific organisations, had a prominent role in information circulation.<br>Individuals with contrasting views on face masks had as much Twitter volume as organisational accounts. |

questions that need to be considered in future research and tweeting practices in pressurised environments, such as an ongoing public health emergency.

## What type of information about face masks was deemed as newsworthy and shareable?

The pandemic, with its global reach, was highly relevant to a wide audience, because the events had such a direct and dramatic impact on individual's daily lives, everyday activities, and behaviours. Consequently, coupled with the high levels of uncertainty that were exacerbated by the contested value of face masks during this period, the need for orientation in this context was acute. Changes in priorities and focus of Twitter users are revealed by the shift in topics and URL subject content (Table 4 and Fig 1) over the timeframes studied (before recommending masks, after recommended and after mandatory). The shift and increasing focus on guidelines/policy in the 'most retweeted tweets' dataset, particularly after they became mandatory in England's and Scotland's shops, suggests that interest in a topic is linked to the latest news and major events–as covered in Agenda-Setting theory, a policy agenda/change has therefore directed the public's attention [52]. As the guideline/policy changes were linked to developments in the scientific evidence, this provides a possible account for the rise in the number of URLs referring to scientific evidence and/or expertise. Additionally, as 'scientific evidence and expertise' was a popular subject category for external URL content, this type of information is clearly circulated on Twitter and is likely to have influenced Twitter users' understanding of science, therefore supports the notion that the science communication landscape is changing through the digitalisation of the information ecology [63].

## When did users choose to share information about face masks during the COVID-19 pandemic?

The overall frequency of tweets relating to face masks reflect how Twitter users' interest on the topic waxed and waned at various points during the pandemic (Fig 2). In particular, peaks in the number of tweets related to face masks further supports our argument that there was an increase in the public's situational awareness as the guidance and policy changed. Similar observations were made for people seeking information about 'face masks' or 'face coverings' on Google Trends [72] i.e. upward trends at the points of guidance/policy change. As these peaks (both tweets and Google searches) only lasted a few days it supports the notion that public attention is short, and people move onto the next topic of interest. It is at this point in time that those implementing policy and supporting the changes need to take advantage of people searching for or sharing information [54–56]. Additionally, the circulation of external resources of information and evidence, as expressed through URL sharing, was more prominent in the early stages of the pandemic, when sensemaking processes were most active [88, 89]. One explanation for this trend is that as face masks were not a cultural norm in the UK, and as the science and guidance was still emergent, this created an information gap; as a result, individuals were more actively engaged in information seeking and sharing practices. In line with studies using Media Dependence Theory as a framework, these observations show that people were demanding information because of the situational ambiguity [59].

While studies have developed theoretical standpoints on how to mitigate the effects of misinformation in public health crises [90], what our findings show is that specific challenges can arise due to the prolonged circulation of out-of-date information, i.e. not strategic misinformation, nor "mis"-information at all. The change in the advice and messaging about face masks is evident in the analysis of URLs circulated referring to scientific evidence and expertise (Tables 6 and 7). Part of the understanding of science is that it is constantly evolving, and this was

provided as the reason for the change in guidance (for instance in Boris Johnson's briefing on the 11 May 2020 [91]). However, the recirculation of articles not recommending use (Fig 3) indicates a lack of public consensus, with some individuals questioning the changes and, in some cases, using out-of-date, and since contested, articles to support their own views [10]. Although some URLs contain an editor's note, the content of a URL may not be read before it is retweeted, with only the headlines captured [92]. As widespread endorsement of myths, conspiracy theories and misinformation that counter the scientific evidence can inhibit preventative behaviours [5, 51, 93–95], this recirculation of out-of-date articles is likely to undermine the changing recommendations and public health messaging and should be countered by official information sources and narratives. How then to contend with this, and how can we 'undo the science' if the messaging that came beforehand e.g. not recommending mask use, is out of date? Should scientific experts and public figures hold back from making statements in fear that, as is often the nature of scientific research, these may become subsequently invalidated? Timing is clearly vital and either the removal or updating out-of-date URLs is paramount, as well as utilising those moments that policy changes and the public are seeking information to reinforce messaging.

## What are the main external resources of information about face masks?

Another layer of complexity to the digital information ecology identified in our analysis is the volume of external resources that Twitter users are exposed to. In line with other studies about Twitter posts and popular domains [46, 47], broadcasting/journalism platforms represented the majority of the domains in both the 'most retweeted tweets' and 'most tweeted URLs' datasets. The results suggest that the exposure/circulation of these newspapers may vary depending on the type of platform used (i.e. Twitter being a specific digital platform) or context (i.e. face masks), as two of the highest ranking newspapers in a separate survey on newspaper reach [87], did not rank as highly in our sample. Additionally, global information sharing practices are evident in our sample as international media outlets, such as the New York Times, are represented. A higher proportion of URLs linked to academic journals/scientific organisations and/or universities was identified when the dataset was filtered to tweets categorised as scientific evidence and expertise, but the twitter volume to these domains was still lower than broadcasting and journalism outlets (Fig 4).

As has been evidenced elsewhere in the literature [52, 96], it is the lack of good quality information that drives uncertainty, and as a consequence spawns uninformed, speculative and erroneous content and people will seek information from elsewhere [97]. One notable example from our dataset of an external resource countering the official government guidance was a domain we categorised as online journalism (which is a blog) ranked 10[th] in the 'most retweeted tweets' domains categorised as 'scientific evidence and expertise'. The content of the identified URLs discouraged the use of face masks. In this case, the domain also had a higher twitter volume (for retweets classified as 'scientific evidence and expertise') than official channels of information such as the UK government and WHO within our sample–therefore corroborating with previous findings that unofficial unregulated resources can be circulated as much as, or more than, official resources [47].

One external official resource that is noticeable missing from our sample and these conversations about scientific evidence and expertise is SAGE's official website which contains (since 29 May 2020 and with a delay [98]) summaries of their meetings and studies feeding into their discussions [99]. Although in some cases, scientific resources such as The Lancet were drawn upon, official government channels of information about the scientific advice were not to the same extent—it is highly likely this is due to accessibility (i.e. readability) of those documents

compared to the overviews/interpretations provided by the media. The dominance of particular domains is also highly likely to be influenced by Twitter users, which we now discuss.

## Which user's tweets were most widely circulated?

As outlined in the existing literature, alongside the diversity of digital platforms, there is also a diversity of communicators within the 'communications fray' [62, 63]. As our study found that the distribution of retweets attached to organisations compared to individuals was 58.16% and 41.84% respectively, we agree with previous critiques that is it vital for Social-Mediated Crisis Communication theory to focus on how multiple actors engage with crisis response [62, 64].

At an organisational level, user profiles associated with broadcasting/journalism organisations had some of the highest interaction rates (Fig 5). In many cases it is likely that they are posting, and therefore amplifying [49], their own news stories, which is why these distributions are also high in the domain categorisation. This dominance corroborates with previous studies which found traditional news media still have a major role in information circulation despite the plurality of voices that are present [43, 63, 65]. Existing research in Social-Mediated Crisis Communication suggests people prefer 'official sources' and encourages the vocalisation from scientists and experts [64], and there are concerns expressed that the plurality of voices can impact the ability of scientific organisations and individuals to gain a high profile [63]. However, there are examples within our sample of this not being the case–for instance, when filtered to 'scientific evidence and expertise' URL content, the Lancet, an academic journal was the user with the highest number of retweets. As with traditional media, it is likely this is why their domain also featured highly.

We also found that individual users, including professional scientists, can trigger a similar level of retweets to organisations (Fig 6). Clearly, if they choose to actively engage with the platform, scientists can gain (but this is unlikely to be in every instance) high levels of exposure. However, at the same time, the dominant involvement of non-professional users (in this context referring to those outside the scientific or medical profession) in the dissemination of scientific information, suggests there is potential for these individuals to also have the same level of exposure as media organisations and those considered to be scientific or academic experts [97]. Previous studies have advised that individuals with potential influence are closely monitored to prevent the spread of misinformation [61]. This leads us to question, as the scientific evidence is not always in consensus i.e., some advocating for face mask use before the guidance changed and others countering the official narrative of the Government when they called for mandating mask use(bearing in mind policy decisions consider scientific evidence alongside other factors [98, 100, 101]), where is the line drawn between it being genuine scientific and/or political debate against harmful content and users being suspended–as was the case for two of the user profiles ranking within the top ten in our analysis?

In sum, the power of Twitter users on the digital information ecology and consequently circulation of scientific information should be acknowledged, and addressed, in the formulation of future public health communication strategies, a recommendation also made in the analysis of Twitter content in previous pandemics [102].

## Conclusions and further work

This article contributes an empirical analysis of information sharing practices on Twitter relating to the use of face masks in the context of COVID-19. Due to the developments in scientific understanding and changes in face mask guidance and policy, we conclude the current literature on digital information ecology is insufficient for capturing the dynamic nature of a long-term ongoing crisis event. Although existing theoretical frameworks have provided useful

approaches for conceptualising user's motivations for accessing information [43, 96], their assessments of source credibility [47, 59] and organisation risk communication strategies [62], our work highlights that there are specific considerations to be factored in when crisis communication is dependent on emergent scientific evidence. We identified that challenges can arise due to the prolonged circulation of out-of-date information, i.e. not strategic misinformation, nor "mis"-information at all, which can have serious ramifications for crisis communication practitioners working to deliver good quality information. Additionally, our analysis shows changes in the frequency of tweets about the topic correspond with key guidance and policy changes. As recommended by previous studies [54, 55], news media exists within an attention economy, these are points in time official channels of information need to utilise the public's information seeking and sharing practices. This is particularly important if out-of-date information is being circulated at the same time.

Our study also has implications for advancing communication research in light of how traditional news media still plays a prominent role within information sharing practices on Twitter despite the plurality of voices present and high volume of external resources being circulated. Concurrently, we found there is scope for academic/scientific/research organisations to have a voice within this landscape. Furthermore, as also identified in previous studies [49, 50], individual people, with strong agendas and/or contrasting viewpoints to one another, can have similar levels of Twitter volume (i.e. number of retweets) and therefore exposure as large organisations. This supports the critique that Crisis-Mediated Communication theory needs to focus on multiple actors in the crisis response [62, 64] and special consideration needs to be given to countering inaccurate information from information sources that are not regulated [61]. Overall, the findings of our study provide implications for advancing crisis communication theory and provide practical implications for crisis communicators and misinformation management.

There are several avenues for further work following this research, including a more granular analysis of the datasets produced, extension of the time analysed, and comparison between devolved nations (our study appears to be more reflective of England's changing guidelines). As we have shown that traditional media and other journalism platforms are so prominent in the circulation of information, further work could inspect whether the scientific evidence and expertise is accurately reflected on these platforms. Furthermore, we have not established if the public discourse impacted media coverage or vice versa [103], further inspection may be able to shed some light on this. This may include identifying echo chambers and/or information cascades [102]. Lastly, we (as have previous studies [52]) discussed that Twitter users are likely to have different motivations for posting and resharing information–however we have not explored these motivations. To truly understand the digital information ecology, including why some posts are shared more than others, and there are differences between people's original tweets and retweeting behaviours, further work on these agendas is welcomed.

## Supporting information

**S1 Appendix. Datasets.** Sheets/tabs show the datasets used and researcher's categorisation of URL subject content, domains and user profiles.
(XLSX)

**S2 Appendix. Codebook and definitions.** Tables display the codes used for manual inductive analysis of text and provides definitions.
(PDF)

**S3 Appendix. Inter-annotator agreement.** Sheets/tabs show the method to calculate the inter-annotator agreement discussed in the paper's methodology and limitation sections. (XLSX)

**S4 Appendix. Analysis overview.** Sheets/tabs show our analysis/provide an overview of the Tweet frequency, topics by timeframe, URL subject categories, domain categories and user profiles. (XLSX)

## Author Contributions

**Conceptualization:** Hannah Baker.

**Data curation:** Hannah Baker, Shauna Concannon.

**Formal analysis:** Hannah Baker, Shauna Concannon.

**Investigation:** Hannah Baker.

**Methodology:** Hannah Baker, Shauna Concannon.

**Software:** Shauna Concannon.

**Supervision:** Emily So.

**Validation:** Shauna Concannon.

**Visualization:** Hannah Baker, Shauna Concannon.

**Writing – original draft:** Hannah Baker, Shauna Concannon.

**Writing – review & editing:** Hannah Baker, Shauna Concannon, Emily So.

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
