## [Decision Letter · Decision Letter 0]

5 Jan 2022

PONE-D-21-31236Information sharing practices during the COVID-19 pandemic: a case study about face masksPLOS ONE

Dear Dr. Baker,

Thank you for submitting your manuscript to PLOS ONE. After careful consideration, we feel that it has merit but does not fully meet PLOS ONE’s publication criteria as it currently stands. Therefore, we invite you to submit a revised version of the manuscript that addresses the points raised during the review process. Please address issues raised by Reviewers and especially focus on research gap, integrate conclusions with earlier research and improve discussion of theoretical and practical contributions. Please submit your revised manuscript by Feb 19 2022 11:59PM. If you will need more time than this to complete your revisions, please reply to this message or contact the journal office at plosone@plos.org. Please include the following items when submitting your revised manuscript:A rebuttal letter that responds to each point raised by the academic editor and reviewer(s). You should upload this letter as a separate file labeled 'Response to Reviewers'.A marked-up copy of your manuscript that highlights changes made to the original version. You should upload this as a separate file labeled 'Revised Manuscript with Track Changes'.An unmarked version of your revised paper without tracked changes. You should upload this as a separate file labeled 'Manuscript'.

We look forward to receiving your revised manuscript.

Kind regards,

Jarosław Jankowski

Academic Editor

PLOS ONE

2. Please provide additional details regarding participant consent. Pease ensure that you have discussed whether  the IRB or ethics committee waived the requirement for informed consent.

3. In your Methods section, please include additional information about your dataset and ensure that you have included a statement specifying whether the collection and analysis method complied with the terms and conditions for the source of the data.

5.In your Data Availability statement, you have not specified where the minimal data set underlying the results described in your manuscript can be found. PLOS defines a study's minimal data set as the underlying data used to reach the conclusions drawn in the manuscript and any additional data required to replicate the reported study findings in their entirety. All PLOS journals require that the minimal data set be made fully available. For more information about our data policy, please see http://journals.plos.org/plosone/s/data-availability.

6. Please note that in order to use the direct billing option the corresponding author must be affiliated with the chosen institute. Please either amend your manuscript to change the affiliation or corresponding author, or email us at plosone@plos.org with a request to remove this option.

Reviewers' comments:

Reviewer's Responses to Questions

**Comments to the Author**

1. Is the manuscript technically sound, and do the data support the conclusions?

Reviewer #1: Yes

Reviewer #2: Yes

Reviewer #3: Yes

2. Has the statistical analysis been performed appropriately and rigorously? 

Reviewer #1: Yes

Reviewer #2: N/A

Reviewer #3: No

3. Have the authors made all data underlying the findings in their manuscript fully available?

Reviewer #1: Yes

Reviewer #2: Yes

Reviewer #3: Yes

4. Is the manuscript presented in an intelligible fashion and written in standard English?

Reviewer #1: Yes

Reviewer #2: Yes

Reviewer #3: Yes

5. Review Comments to the Author

Reviewer #1: Dear Author(s)

Thank you for the opportunity to read this interesting piece of work. The manuscript is well-written. However, I feel the manuscript is rather long and descriptive falling short of critically engaging with previous research. three major issues need to be addressed. First, the research gap your study attempts to address needs to be expressed more explicitly. Second, the conclusions section failed to engage with previous research. Do your study findings confirm/refute previous research? Finally, the manuscript falls short of discussing the theoretical and practical contributions/implications of study.

Good luck with revising your manuscript.

Reviewer #2: The manuscript presents a very interesting topic, of global scope and likely to interest the world of academia, including sociologists, medical staff, marketers and social media experts, but also people not associated with the scientific community (eg., influencers). It has a universal character, which, however, is not a disadvantage, but deserves recognition. It is not written at a high level of generality, on the contrary, the article is very carefully prepared. The manuscript is presented in an intelligible fashion: the language is clear, correct, and unambiguous. No typographical or grammatical errors were found. The research was well thought out and well designed. The manuscript describes a technically sound piece of scientific research with data that supports the conclusions. The authors have fully provided all data on which the conclusions in the manuscript are based. These data have been provided both within the manuscript and its supplementary information.

Reviewer #3: Review of

Information sharing practices during the COVID-19 pandemic: a case study about face masks

Submitted to PLOS ONE

PONE-D-21- 31236

The paper investigates the sharing of information about face masks during the COVID-19 pandamic in the UK.

In a general sense, there is much to like about the paper. The paper is well written, and the overall argumentation is well crafted.

Nevertheless, I have three substantial issues with respect to the paper that limit the scientific contribution substantially.

MAJOR ISSUES

(1) CONTRIBUTION:

a. I think that you have an interesting story. However, I was not really convinced about the contribution of the paper. I believe your findings are important, but I really suggest coming up already in the introduction with the main findings. It takes much too long to see the results.

b. When reading the abstract, I was wondering about findings and what I can learn from this. You need to be much more specific to “sell” your results.

c. I also suggest focusing on findings that are (a) new and (b) impactful. For example., I think that it was interesting to discuss the relevance of outdated scientific results (page 32 in the manuscript). This is a key finding! I have no problem if you decide shortening the paper and focus on less but interesting findings.

d. Many findings that you report are not so surprising and lack (causal) support. You report sharing behavior, but you do not control for many drivers that might lead to specific sharing behavior (e.g., discussion of a news issue on TV, sharing of content by a celebrity etc.). Thus, I wonder if you should reduce the storyline and focus on the really new insights.

e. Please add a new table that shows in a top-down fashion an overview of all your studies (data, method, and the subsequent findings).

(2) THEORY: The paper is basically free of theory. While I do not believe that you must add theory to the paper, I suggest presenting insights that are either supported by theory or that substantially challenge theories. Currently, I think your findings are in line with agenda setting theories. Thus, what can we learn? If we cannot learn much for theory testing and building, then I suggest focusing on the managerial impact of your results.

(3) ROBUSTNESS:

a. I would like to see your findings contrasted to google trends.

b. Please also show your trend over time and clearly discuss the events that occurred during that time. This allows the reader to better understand confounding events that might drive search (via Google) or tweeting.

c. Please report the reach of the newspapers when discussing the popular domains.

d. I think it would also be interesting to discuss why some other relevant posts have not been shared as much (discussion of non-findings).

Overall, I am supportive towards this paper – but the authors need to streamline the paper to better show their contribution.

I thank you for the opportunity to read this manuscript and I hope that the authors will find the comments helpful and that they will continue to follow this research stream.

6. PLOS authors have the option to publish the peer review history of their article (what does this mean?). If published, this will include your full peer review and any attached files.

Reviewer #1: No

Reviewer #2: No

Reviewer #3: No

---

## [Author Response · Author response to Decision Letter 0]

25 Feb 2022

We thank the reviewers for the time spent reviewing our manuscript, and we have edited the manuscript to address the points raised. This includes (but is not limited to): ensuring that we set a theoretical context, 2) making our theoretical and practical contributions clearer, 3) shortening the paper to focus on our key findings.

A detailed breakdown of our responses is available in the following document attached to this submission: 'Response to reviewers.pdf'

In this document our responses are shown in blue text. We have also provided a copy of the revised manuscript with tracked changes on the submission system. 

We feel that these changes have significantly improved the paper, so once again thank you for the 

time spent to date in the review process. We hope that the manuscript is now ready for publication in 

PLOS ONE.

---

## [Decision Letter · Decision Letter 1]

21 Apr 2022

Information sharing practices during the COVID-19 pandemic: a case study about face masks

PONE-D-21-31236R1

Dear Dr. Baker,

We’re pleased to inform you that your manuscript has been judged scientifically suitable for publication and will be formally accepted for publication once it meets all outstanding technical requirements.

Kind regards,

Chi Ho Yeung

Academic Editor

PLOS ONE

Additional Editor Comments (optional):

Reviewers' comments:

Reviewer's Responses to Questions

**Comments to the Author**

1. If the authors have adequately addressed your comments raised in a previous round of review and you feel that this manuscript is now acceptable for publication, you may indicate that here to bypass the “Comments to the Author” section, enter your conflict of interest statement in the “Confidential to Editor” section, and submit your "Accept" recommendation.

Reviewer #1: All comments have been addressed

Reviewer #2: All comments have been addressed

Reviewer #3: All comments have been addressed

2. Is the manuscript technically sound, and do the data support the conclusions?

Reviewer #1: Yes

Reviewer #2: (No Response)

Reviewer #3: Yes

3. Has the statistical analysis been performed appropriately and rigorously? 

Reviewer #1: Yes

Reviewer #2: (No Response)

Reviewer #3: Yes

4. Have the authors made all data underlying the findings in their manuscript fully available?

Reviewer #1: Yes

Reviewer #2: Yes

Reviewer #3: Yes

5. Is the manuscript presented in an intelligible fashion and written in standard English?

Reviewer #1: Yes

Reviewer #2: Yes

Reviewer #3: Yes

6. Review Comments to the Author

Reviewer #1: Dear Authors

Thank you for addressing the concerns I have raised in the review report.

The issues were addressed in full and I, therefore, recommend accepting your manuscript for publication.

Good luck

Reviewer #2: (No Response)

Reviewer #3: (No Response)

7. PLOS authors have the option to publish the peer review history of their article (what does this mean?). If published, this will include your full peer review and any attached files.

Reviewer #1: No

Reviewer #2: No

Reviewer #3: No

---

## [Editor Report · Acceptance letter]

26 Apr 2022

PONE-D-21-31236R1 

Information sharing practices during the COVID-19 pandemic: a case study about face masks 

Dear Dr. Baker:

I'm pleased to inform you that your manuscript has been deemed suitable for publication in PLOS ONE. Congratulations! Your manuscript is now with our production department. 

Kind regards, 

on behalf of

Dr. Chi Ho Yeung 

Academic Editor

PLOS ONE